# Evaluation of Effective Composite Biosorbents Based on Wood Sawdust and Natural Clay for Heavy Metals Removal from Water

**DOI:** 10.3390/ma16155322

**Published:** 2023-07-28

**Authors:** Roberta Del Sole, Alena A. Fogel, Vladimir A. Somin, Giuseppe Vasapollo, Lucia Mergola

**Affiliations:** 1Department of Engineering for Innovation, University of Salento, via per Monteroni Km 1, 73100 Lecce, Italy; giuseppe.vasapollo@unisalento.it (G.V.); lucia.mergola@unisalento.it (L.M.); 2Humanitarian Institute, Higher School of Jurisprudence and Forensic Technical Expertise, Peter the Great St. Petersburg Polytechnic University, Politekhnicheskaya St., 29, 195251 Saint Petersburg, Russia; alena.fogel22@gmail.com; 3Institute of Biotechnology, Food and Chemical Engineering, Polzunov Altai State Technical University, Lenina Avenue, 46, 656038 Barnaul, Russia; vladimir_somin@mail.ru

**Keywords:** adsorption, bentonitic clay, wood sawdust, copper ions removal, nickel ions removal, kinetic study

## Abstract

Bentonitic clay and wood sawdust are natural materials widely available in nature at low cost with high heavy metals sorption properties that, in this work, were combined to achieve an effective composite biosorbent with high sorption properties and enhanced mechanical stability. Pine, aspen, and birch wood sawdust, as well as different bentonite clays and different sawdust modification methods (H_3_PO_4_ or HCl) were used for preparing new composite biosorbents. A mixture of wood sawdust and bentonite in a ratio of 2:1 was used. All materials were characterized by using X-ray diffraction (XRD), Fourier transform infrared spectroscopy (FTIR), and scanning electron microscope (SEM) methods and tested for Cu and Ni ions removal from water. The adsorption process for all composite biosorbents was well described from a pseudo-second order kinetic model (R^2^ > 0.9999) with a very high initial adsorption rate of Cu and Ni ions and a maximum uptake recorded within 2 h. The results have shown that the adsorption capacity depends mainly on the kind of wood and the acid treatment of the wood that enhances the adsorption capacity. At a concentration of 50 mg/L, the biosorbent prepared using birch wood sawdust showed the worst performance, removing barely 30% of Cu and Ni ions, while aspen wood sawdust improved the adsorption of Cu (88.6%) and Ni (52.4%) ions. Finally, composite biosorbent with pine wood sawdust showed the best adsorption be haviour with an efficiency removal of 98.2 and 96.3% of Cu and Ni ions, respectively, making it a good candidate as an inexpensive and effective biosorbent for the removal of heavy metals.

## 1. Introduction

Heavy metal water contamination from industrial wastewater is a relevant environmental problem. Many industries, such as automotive, metal finishing, electroplating, mining, electric cable manufacturing, tannery, steel, textile, and petroleum refining and rubber processing industries, discharge into ecosystems and water bodies different concentrations of organic and inorganic pollutants such as dyes [1], pesticides [2], and fluorides [3,4] and heavy metals like copper, lead, mercury, cadmium, zinc, uranium [5], tungsten [6], cobalt, and nickel [7]. Cu and Ni ions are the most common heavy metals in the above wastewaters [8,9].

Heavy metals are hazardous for living organisms. Being dangerous substances, they can also form extremely toxic compounds, interacting with other substances. They do not change into safe compounds in the metabolism. In the food chain, they remain present in water, plants, animals, and humans, and in quantities hundreds and thousands of times exceeding their content in the natural environment [10,11]. Cu is considered one of the most significant heavy metals utilized in electroplating industries, as it induces severe toxicological consequences at high dosages. After uptake, it is accumulated in the brain, skin, liver, pancreas, and may cause heart disease [12]. Drinking water with higher-than-normal levels amount of Cu may induce nausea, vomiting, stomach cramps, or diarrhoea. High intakes of Cu can cause liver and kidney damage and even death [13]. According to the World Health Organization (WHO, the year 2020), the nickel concentration in drinking water is 2–13 μg/L in Europe. Nickel is potentially dangerous due to its carcinogenic properties for human beings. Indeed, it can cause several problems such as cardiac arrest, kidney damage, and respiratory and gastrointestinal disorders [14,15]. Due to the risk for living beings and mainly for humans of the above-described toxic metals, finding a suitable solution is essential in terms completely removing them or by decreasing them to a low level in water. There are many treatment processes that can be used for metal ions removal from various industrial effluents: precipitation, flocculation, membrane separation, ion exchange, biosorption, adsorption, evaporation, and electrolysis [16,17,18]. The most commonly adopted approaches are adsorption, ion exchange, chemical settling, and reverse osmosis, while adsorption has gained significative consideration due to the high efficiency demonstrated in heavy metal removal and several different adsorbents have been used in adsorption methods. However, their high cost causes restrictions in use. Therefore, the use of low-cost adsorbents for wastewater treatment is recommended at present due to their local availability, technical feasibility, engineering applicability, and cost effectiveness. For this reason, many studies have been carried out to find effective and low-cost adsorbents [19]. Different natural adsorbents [20] have been reported for removal of copper and nickel such as chitosan [21,22,23], corncob [24], jute fibre [25], cotton [26], fruit peel [27], sawdust [7,10,16,28], cellulose [29], bentonite [30,31,32], etc.

Bentonitic clay is a natural inorganic material with low cost and high-sorption properties that is being studied as a heavy metal adsorbent. Bentonite contains silicate layers of 2:1 with a surface negatively charged for the substitution of aluminium and magnesium for silicon and aluminium in a tetrahedral and octahedral layer, respectively. The main constituent of bentonite is montmorillonite, a 2:1 mineral with one octahedral sheet and two silica sheets, which is responsible for heavy metals retention and occurs mainly through a cation exchange process [33]. Moreover, the great specific surface area, the excellent physical and chemical stability, and several other structural and surface properties of bentonite represent other important advantages for the use of clay in adsorbent materials [30,34,35,36].

Traditionally, bentonitic clay is measured out in clear water, mixed and directed to sediment bowls for sedimentation. After that, the clarified water is purified using filters. The existence of such technology explains the hydrophilicity of bentonitic clays. In this context, they create highly stable suspensions that require prolonged thickening. To hasten this process, water can be acidified in a separating tank. However, this approach is unprofitable and degrades the ion exchange properties of bentonite. In this connection, the idea of receiving new sorption material, which can be used as a filtering medium, looks interesting. Application of this material will allow for a reduction in treatment plant area and considerably reduce the price of clearing water of heavy metal ions. Bentonite can offer a high porosity of the layer, ensuring effective and prolonged filtering. Various kinds of fibres, such as minerals and plants, and other natural materials can be used. For example, in a work by Mamyachenkov et al., it was proposed to use mineral wool together with bentonitic clay [37].

To improve its applicability for heavy metals removal, clay minerals modification or functionalization with suitable organic compounds is desirable. For instance, when considering organic matter containing functional groups that enable uptake of metal ions (e.g., polymer, silane coupling agent), researchers have explored the use of such modifiers to investigate various functionalized organoclays [38]. Among them, at the present time, the use of wood sawdust for sorption of heavy metals in effluents has been an object of study in a lot of research [15,39,40,41]. This is a natural material available in large quantities and, being made of waste products, has a low price. Wood sawdust can be used as an adsorbent of copper and nickel, largely due to its lignocellulosic composition. Cellulose and lignin, being part of the structure, possess hydroxyl, carboxylic, and phenolic groups that facilitate the binding of metal cations. Furthermore, wood sawdust is widely available and has good mechanical stability as well as some other advantages [28]. Various ways of processing such as mechanical, physical, chemical, and physicochemical, including thermal processing of raw materials, are applied to increase adsorption capacity of biosorbents. Chemical treatment can be carried out by acid, alkali, and chlorite solutions, facilitating delignification, bleaching, and removal of hemicellulose and other impurities present in the biomaterial [29].

This work aims to combine for the first time the natural adsorption properties of wood sawdust and bentonitic clay to achieve an effective heavy metal sorption material with high mechanical stability and good sorption properties for water treatment. In detail pine, aspen, and birch wood sawdust, different bentonite clays and different sawdust modification methods were used to prepare composite biosorbents and their sorption properties were compared through their application for removal of Cu and Ni ions from water in batch experiments. Composite biosorbents characterization by using of X-ray diffraction (XRD), Fourier transform infrared spectroscopy (FTIR), scanning electron microscope (SEM), and energy dispersive X-ray (EDX) methods was made. To investigate the adsorption mechanism of metal ions onto the sorbent, four kinetic models were analysed: first order reaction model, pseudo-first order equation, second-order reaction model, and pseudo-second order reaction model.

## 2. Materials and Methods

### 2.1. Materials

Copper sulphate (CuSO_4_) (≥98.0%) and nickel sulphate (NiSO_4_) (≥99.0%) were purchased from Sigma-Aldrich (Steinheim, Germany). H_3_PO_4_ (>85% in water), HCl (37% in water), Na_2_CO_3_ (≥99%), provided from Sigma-Aldrich (Steinheim, Germany), were chemically of pure quality and used without further purification. Elemental standard solutions of Cu(II) and Ni(II) for ICP-AES analysis were prepared by appropriate dilution of 1000 mg/L stocks purchased from Fluka. Nitric acid (67–69% in water) for trace metal analysis was supplied from Romil-Spa (Prato, Italy). All solutions were prepared using ultrapure water obtained from a water purification system (Human Corporation, Seoul, Republic of Korea).

### 2.2. Composite Biosorbents Preparation

Two different materials were used for biosorbents preparation: wood sawdust and mineral clay. In detail, three different types of wood sawdust were used: pine, aspen, and birch, and two different types of bentonitic mineral clay: Ca-bentonite obtained from Taganskoe deposit, East Kazakhstan, and Khakass bentonite from The Tenth Farm deposit, Chernogorsk, Russia.

The sawdust was sieved into 20–50 meshes (0.20–0.50 mm) and it was treated with H_3_PO_4_ or HCl. A total of 10 g of sawdust was mixed with 100 mL 5% H_3_PO_4_ or 0.5 N HCl and kept 24 h at room temperature. The modified wood was washed with distilled water to remove residual materials (pH = 5–6). The rinsed sawdust was oven-dried at 120 °C until a constant weight was reached.

The bentonitic clay was activated with sodium carbonate to increase and make more uniform its porous structure, changing the chemical nature of the surface and increasing its sorption capacities. An amount of 10 g of bentonite was mixed with 100 mL 5% Na_2_CO_3_ and kept 24 h at room temperature. The suspension was filtered, the residue was washed with distilled water, and then dried in the oven at 120 °C until a constant weight was reached.

To prepare composite biosorbent materials (Table 1), the modified sawdust was mixed with bentonitic clay in a ratio of sawdust:bentonite of 2:1, then the composite was dried, crushed, and heat treated at 150 °C for 2.5 h.

### 2.3. Samples Characterization

Wood sawdust, bentonitic clay, and composite biosorbents were examined by using an X-ray powder diffractometer (Rigaku Ultima+) equipped with Cu Kα radiation with a wavelength of 0.154 nm. XRD spectra were acquired at room temperature over the 2θ range of 0–50°.

Functional groups were determined by the Fourier transform infrared (FTIR) spectroscopy. Spectra were collected with a spectrometer using KBr pellets and recorded in the range of 4000–500 cm^−1^ by using a JASCO 660 Plus infrared spectrometer (Jasco, Palo Alto, CA, USA). KBr pellets of samples were prepared by mixing 1 mg of each material with 150 mg KBr and 64 scans were taken per sample.

The morphology of the surface of composite biosorbents was analysed by using a ZEISS EVO 40 scanning electron microscope (SEM) in high vacuum mode. Elemental analysis was performed by using SEM instrument equipped with energy-dispersive X-ray (EDX) Bruker 127 eV model XFlash detector 5010 (Bruker, Bremen, Germany). Sample preparation was carried out by applying a small amount to conductive carbon tape.

### 2.4. Adsorption Experiments

For studying the sorption capacity of the prepared composite biosorbents in static conditions, 100 mL of standard solutions with an initial concentration of Cu or Ni ions of 50 mg/L and 10 mg/L were prepared. To each solution, 1 g of sorbent was added and batch experiments were carried out at 25 °C in a mechanical shaker for several hours. After shaking, the solution was filtered through Millipore filters. The metal ion concentration, before and after the adsorption experiments, was determined by using inductive coupled plasma optical emission spectroscopy (ICP-OES) iCAP 6000 Series (Thermo Scientific, Waltham, MA, USA, http://www.thermoscientific.com, accessed on 1 July 2023).

Adsorption capacity *q* (mg/g) was calculated as follows:(1)$$q=\frac{\left(C_{0}-C_{e}\right)}{m\;1000}\;V$$
where *C*_0_, *C_e_* are, respectively, initial and equilibrium concentration of Cu or Ni ions in solution (mg/L), *V* volume test (mL), *m* weight of sorbent (g).

For kinetics study, 1 g of composite biosorbent was added in standard solutions of Cu or Ni ions and kept from 0.5 to 48 h and the concentration of Cu or Ni ions in the solution was monitored at fixed time intervals during this period:(2)$$q_{t}=\frac{\left(C_{0}-C_{t}\right)}{m\;1000}\;V$$
where *C*_0_, *C_t_* are, respectively, initial concentration and concentration at time t of Cu or Ni ions in the solution (mg/L), *V* volume test (mL), *m* weight of sorbent (g).

### 2.5. Kinetic Studies

The kinetics of the adsorption process were studied by carrying out a set of adsorption experiments at 25 °C and monitoring the amount adsorbed at regular intervals as previously described. The adsorption kinetics commonly involves two stages: a rapid removal step followed by a much slower step before the equilibrium is reached. In first order kinetics, the rate of the adsorptive interactions can be obtained with the following equation [31,42]:(3)$$-\frac{dC_{t}}{dt}=K_{1}C_{e}$$
where *C_e_* and *C_t_* are the concentration of Cu or Ni ions in solution at equilibrium and at time *t* (mg/L), respectively, and *K*_1_ is the first order adsorption rate constant.

Integrating Equation (3) for the boundary conditions *t* = 0 to *t* = *t* and *C_t_* = *C*_0_ to *C_t_* = *C_t_* gives the form:(4)$$lnC_{t}=lnC_{0}-K_{1}t$$

So, this equation shows that a plot of *lnC_t_* versus *t* indicates the linear plot of first order equation.

The pseudo-first order kinetic model based on the solid capacity for sorption analysis is:(5)$$-\frac{dq_{t}}{dt}=K_{s1}(q_{e}{-q}_{t})$$
where *q_e_* and *q_t_* are the values of amount adsorbed per unit mass at equilibrium and at time *t*, respectively, and *K_S_*_1_ is the pseudo-first order adsorption rate constant.

Integrating Equation (5) for the boundary conditions *t* = 0 to *t* = *t* and *q_t_* = 0 to *q_t_* = *q_t_* the linearized form is obtained and described by Equation (6):(6)$$ln\;{(q}_{e}{-q}_{t})=ln\;q_{t}-K_{s1}\;t$$

Therefore, from Equation (6), it can be deduced that a linear plot of *ln*(*q_e_ − q_t_*) versus *t* indicates the reaction is a pseudo-first order.

Second-order rate equation is widely used for metal ion sorption, which can be represented as:(7)$$-\frac{dC_{t}}{dt}=K_{2}C_{t}^{2}$$

Rearranging Equation (7) and integrating for the boundary conditions *t* = 0 to *t* = *t* and *C_t_* = *C*_0_ to *C_t_* = *C_t_* as shown by:(8)$$\frac{1}{C_{t}}-\frac{1}{C_{0}}=K_{2}t$$

So, Equation (8) shows a linear plot of 1/*C_t_* versus *t* indicating that the reaction rate is of second order.

A pseudo-second order rate expression based on the sorption equilibrium capacity may be represented as:(9)$$\frac{dq_{t}}{dt}=K_{S2}{{(q}_{e}{-q}_{t})}^{2}$$

Rearranging Equation (9) and integrating the obtained equation for the boundary conditions *t* = 0 to *t* = *t* and *q_t_* = 0 to *q_t_* = *q_t_* gives:(10)$$\frac{1}{q_{e}-q_{t}}=\frac{1}{q_{e}}+K_{S2}\;t$$
which is the integrated rate law for a pseudo-second order reaction.

Rearranging Equation (10):(11)$$\frac{t}{q_{t}}=\frac{1}{h}+\frac{1}{q_{e}}\;t$$
where *h* = *K_s2·_q_e_^2^*, Equation (11) shows a linear plot of *t*/*q_t_* versus *t* confirming the reaction rate has a pseudo-second order kinetic.

## 3. Results

### 3.1. Batch Study of Cu and Ni Ions on Composite Biosorbents

Firstly, Taganskoe Ca-bentonite modified with different kinds of wood (pine, aspen, and birch) treated with H_3_PO_4_ were prepared (see Figure 1) and batch experiments were performed to evaluate their sorption capacity and the ion removal percentage (Table 2) with an initial concentration of 50 mg/L of Cu or Ni ions. Results have shown the higher sorption capacity of the composite material of 4.9 and 5.2 mg/g for Cu and Ni ions, respectively, with an efficiency of extraction from the solution of 98.2 and 96.3%, respectively, when Taganskoe Ca-bentonite and pine wood sawdust were used (t_1_Pine-TB). Alternatively, t_1_Birch-TB, only practically removes almost 30% of Cu or Ni ions from the solution. Thus, it can be observed that the kind of wood significantly affects the performances of the composite material. As can be seen in Table 2, pine wood showed the best behaviour with similar values for both ion metals, while aspen (t_1_Aspen-TB) gave different results for Cu (88.6%) and Ni (52.4%) ions.

Starting from this first evidence, pine sawdust was selected for further studies. In detail, the t_1_Pine-TB was compared with two kinds of sawdust treatments (H_3_PO_4_ and HCl). Moreover, the influence of clays of different deposits was studied. The results of sorption capacity and removal percentage on studying the above biosorbents (C_0_ = 50 mg/L of Cu and Ni ions) are reported in Table 3. It is evident that the nature of clay essentially does not influence the sorption properties of the material. Efficiency of extraction of Cu and Ni ions from the solution is very similar for materials with one kind of wood (pine) and different kinds of bentonite: Taganskoe Ca-bentonite and Khakass bentonite give 98.2 and 96.3% for Cu ions, respectively. On the other hand, the use of a sawdust treatment is important for increasing the sorption properties of the sorbent, while the strength of the acid (H_3_PO_4_ or HCl) does not significantly affect the adsorption performances, especially for Ni ions. Indeed, the preliminary acid treatment of wood sawdust considerably influences uptake performance. This can be seen when comparing Cu ions removal from t_1_Pine-TB (98.2%) treated with H_3_PO_4_ and from untreated nPine-TB (73.8%). Meanwhile, the percentage of Ni ions removal was 84.1% for t_1_Pine-KB and 83.9% for t_2_Pine-KB treated with H_3_PO_4_ and HCl, respectively.

From the results summarized in Table 3, it can be concluded that the kind of clay does not significantly affect the results, while the sawdust acid treatment is important. Finally, the strength of the acid seems not to be important compared to H_3_PO_4_ and HCl treatment. From these results, it is evident that materials based on Taganskoe deposit Ca-bentonite and acid-treated pine wood sawdust showed the best sorption performance.

### 3.2. Characterization of t_1_Pine-TB

Starting from the results obtained in the previous experiments, t_1_Pine-TB was subjected to further characterization studies. The biosorbents’ pH was controlled during the preparation process giving pH values in water for all the prepared composite biosorbents of around 6. The XRD diffraction patterns of t_1_Pine-TB and the corresponding starting materials were compared, and the results are reported in Figure 1. Changes in position and intensity of the peaks for these samples were investigated. Sorption material t_1_Pine-TB showed the presence of peaks typical for wood and clay, but with a lower number and intensity compared to its starting materials. The position of peaks in the 2θ range at 15° and 27° were related with wood sawdust and bentonitic clay, respectively. In the range of 30–50°, the number of peaks for t_1_Pine-TB was reduced compared to bentonite clay. Characteristic widening of the XRD peaks and a decrease in intensity are commonly observed. The incorporation of materials mixed with bentonite and wood sawdust expanded the basal spacing in a 2θ range of 7° and 20°, which is typical for montmorillonite. This was accompanied by a change in intensity. The diffraction peak of bentonite at 21° increased after intercalation with wood sawdust and a peak of 7.58° are also characteristic of the change in position for the material, as compared with bentonite with an average of 1°. The decrease in the intensity of the peak and its disappearance indicate the formation of an intercalated–exfoliated structure in composites. This causes changes in the regular pattern of the clay structure and a partial destruction of the original structure [43]. t_1_Pine-TB was also obtained at a peak of around 23°, which is characteristic of cellulose components in wood. According to the results, it can be concluded that t_1_Pine-TB biosorbent XRD relates to a partial destruction of the crystalline structure of bentonite. The XRD measurements showed a change in the intensity, position, and number of peaks related to the change in structural physical properties of starting materials (wood and clay).

The FTIR spectrum of materials based on Taganskoe bentonite and different kinds of wood sawdust (t_1_Pine-TB, t_1_Aspen-TB, and t_1_Birch-TB) are shown in Figure 2, where the region of 1800–700 cm^−1^ containing the most useful signals is considered. A complete collection of bands positions and related function groups for the three kinds of materials are described in Table 4.

Each spectrum shown in Figure 2 is complex and characterized by several signals assigned to main wood and bentonitic clay parts. Peaks centred at 1595, 1510/1507, 1270, and 1250 cm^−1^ are ascribable to characteristic bending or stretching modes of some typical groups in lignin. Peaks at 1470, 1425, 1385, 1160/1165, 1111, and 1040 cm^−1^ are assigned to characteristic bending or stretching vibrations of different groups for lignin and cellulose. Peaks at 1739 cm^−1^, and in the range of 1321–1317 cm^−1^, are ascribable to characteristic bending or stretching modes of different groups in cellulose. The main typical bands are roughly at 1640, 1510, and 1040 cm^−1^ and are assigned to the stretching vibrations of C=O, aromatic and C–O [44]. However, in all samples with different kinds of wood, different positions and relative intensity of the peaks are found. For the material with pine wood, the absence of the peak at 1595 cm^−1^ is noticeable, and there is a higher band intensity at 1510 and 1030 cm^−1^ compared to pine. For materials with birch and aspen woods, the absence of peak 1270 cm^−1^ and the band intensity at 1470 and 1111 cm^−1^ are greater. Moreover, the peak at 1739 cm^−l^ appears considerably changed in the spectrum of material with pine wood compared to birch and aspen woods. In the sorbent with birch and aspen woods, this peak is intense, while in the sorbent with pine wood, it decreases becoming a shoulder.

It is possible to assume that positions and relative intensity of peaks in 900–600 cm^−1^ region are connected with the presence as a part of the bentonitic clays. For this spectral region, Si–O and Al–O deformation are characteristic. Judging by Figure 2, for the material with birch sawdust, there are no peaks at 800 and 795 cm^−1^. Also, in all the spectra, we can see peaks at 1645/1640 cm^−1^ and 1040 cm^−1^, which is typical for clay.

To evaluate how a different acidic treatment can affect the final composites, the FTIR spectra of composite biosorbents t_1_Pine-KB and t_2_Pine-KB are analysed and their spectra are shown in Figure 3. Spectra are similar with slight distinctions observed only in intensity of bands. Thus, we can suppose the chemical composition of biosorbents remains relatively unchanged, regardless of whether acid (H_3_PO_4_ or HCl) is applied in the sawdust treatments. This result justifies the results obtained in the adsorption studies.

The surface morphology of all composite sorbents prepared was investigated by SEM and the results are reported in Figure 4. As can be seen, adsorbents had an irregular and porous surface. The microphotographs of t_1_Birch-TB (Figure 4c) show a harder structure and bigger, more symmetric pores. The microphotographs of t_1_Pine-TB and t_1_Aspen-TB (Figure 4a,b) clearly reveal the presence of asymmetric pores and an open-pore structure, which might be the reason for high metal ions adsorption due to a high internal surface area being provided. A distribution of the bentonitic clay throughout the whole surface of clay particles can be observed. Bentonite covers the surface of sawdust with a thin film.

Chemical composition of t_1_Pine-TB was studied using elemental EDX analysis. The EDX carried out at two different points of the surface of sorbent is shown in Figure 5a,b. Chemical composition changes on the different points of the surface should be observed. In Figure 5b, a higher content of bentonite characterized by a high content of silicon and aluminium, typical of clay composition, was found. In addition, sodium likely plays a role in cation exchange in the clay, and its content affects the sorption properties of materials. Generally, sorbents with a higher content of sodium possess higher sorption capacity [45,46].

### 3.3. Kinetic Studies

Important information regarding the contact time of the adsorbate on the adsorbent composite and the reaction coefficients can be obtained from kinetic study. Here, kinetics of adsorption properties of t_1_Pine-TB that showed best sorption capacity were investigated.

Figure 6 shows the percentage of Cu ions removal at two different initial concentrations (50 and 10 mg/L) versus the times of t_1_Pine-TB based on Taganskoe bentonite and pine wood modified with H_3_PO_4_. For both concentrations of the metal ions in the solution, adsorption was very fast initially. For the concentration C_0_ = 50 mg/L (Figure 6a), the maximum uptake is recorded within 2 h (96%).

After 24 h, adsorption was slightly reduced from 96% to 95%. For the initial concentration of C_0_ = 10 mg/L (Figure 6b), the rate of adsorption gradually increased up to 95% and 97% in 2 and 24 h, respectively.

To investigate the adsorption mechanism involved, four kinetic models were analysed: first order reaction model related to the solution concentration, pseudo-first order equation related to the solid capacity, second-order reaction model related to the solution concentration, and pseudo-second order reaction model related to the solid phase sorption.

Figure 7 shows a plot of the linearized order reaction kinetics of Cu ions onto t_1_Pine-TB at various initial concentrations of Cu ions. The adsorption rate has been described first with the first order mechanism. Linearized form of the first order plots is shown in Figure 7a. The linear fit (R^2^ = 0.7044 for initial concentration of 10 mg/L) indicates that the kinetic does not follow a first order reaction model as well as for a concentration of 50 mg/L (correlation coefficient R^2^ = 0.3432). Figure 7b shows the linear plot of the pseudo-first order model. Plots have poor correlation coefficients (0.0433 for C_0_ = 10 mg/L and 0.3398 for C_0_ = 50 mg/L) which indicated that the adsorption did not follow a pseudo-first order reaction. A plot of the linearized form of the second-order reaction models is shown in Figure 7c. The linear fit (correlation coefficient, R^2^ = 0.8058 for C_0_ = 10 mg/L and 0.2825 for C_0_ = 50 mg/L) indicates that the second-order reaction model is more suitable for low concentrations (10 mg/L) than for high concentrations (50 mg/L). Pseudo-second order kinetic plots (Figure 7d) had the best linearity (correlation coefficient, R^2^ = 0.9999 for C_0_ = 50 mg/L and 0.9997 for C_0_ = 10 mg/L) and appeared to give a better understanding of the interactions.

So, it can be concluded that the pseudo-second order kinetic model better describes Cu ions adsorption over the whole period according to the correlation coefficients, which is higher than correlation coefficients of other order reactions models and also in agreement with other published results for sawdust- or clay-based sorption systems [7,15,31,41].

Although linear approximation of kinetic dependences was chosen as the most widely used in the description of sorption processes and allowed a fairly good description of the sorption kinetics, the nonlinear pseudo-second order was also performed by using the following Equation (12):(12)$$q_{t}=\frac{K_{2}q_{e}^{2}t}{{1+K_{2}q}_{e}t}$$
where *K_2_* is the pseudo-second order constant and *q_t_* and *q_e_* are the amount (mg) of metal ion adsorbed per unit mass of the adsorbent (g) at time t and at equilibrium, respectively. For experiments with Cu ions (C_0_ = 50 mg/L), nonlinear Equation (12) gave values of *q_e_* and R^2^ equal to 4.92 mg/g and 0.999, respectively, while at C_0_ = 10 mg/L values of *q_e_* and R^2^ equal to 1.06 mg/g and 0.800, respectively.

In the same way as Cu ions, Ni ions were tested for kinetic studies on t_1_Pine-TB. Figure 8 shows a similar percent of Ni ions removal of those described for Cu ions. The rate of adsorption was initially very fast. It can be noted that almost 95% removal of Ni ions for *C*_0_ = 50 mg/L and 95% for *C*_0_ = 10 mg/L took place within 1 h and then gradually increased up to 97% and 98%, respectively. The observed quick sorption rate can suggest that the sorption process is manly film-diffusion controlled, and this may be caused by the relevant difference in the concentration of bulk solution and solid phase.

Also, for Ni ions, four kinetic models were investigated. Figure 9 shows a plot of the linearized order reaction kinetics of nickel ions on t_1_Pine-TB biosorbent. Comparing the correlation coefficients for all order reaction models of adsorption R^2^, pseudo-first, and second-order models reveal very poor values of 0.3388, 0.0065, and 0.3081, respectively, when *C*_0_ is 10 mg/L and R^2^ equal to 0.0562, 0.5214, and 0.0990, respectively, and when *C*_0_ is 50 mg/L. Similarly to Cu ions, only the pseudo-second order model is characterized by high R^2^ values (1.0 for *C*_0_ = 10 mg/l and 0.9999 for *C*_0_ = 50 mg/L), and it is possible to use this model for the description of the sorption process.

Also, for Ni ions, nonlinear pseudo-second order Equation (12) was tested, giving similar results to those of Cu ions and confirming the convenience of the linearization approach.

Therefore, we can conclude that the pseudo second-order kinetic model provides the best correlation for all the systems studied. In addition, the initial adsorption rates for the metal ions were higher at lower initial metal ion concentrations. So, the rate is dependent on the initial concentration.

A comparison of the sorption capacity of heavy metals on similar biosorbents was shown in Table 5.

It is evident that the removal capacity of the t_1_Pine-TB towards Cu and Ni ions is comparable to other available adsorbents.

We also verified the behaviour on a different heavy metal ion by monitoring the sorption capacity of t_1_Pine-TB biosorbent for chromium(VI) ions. The sorption capacity was about 7.5 mg/g. A preliminary evaluation on the capability of the biosorbent to extract oil products was also made. The sorption capacity in this last case was about 3.6 mg/g and it confirms the large sorption potential of the developed t_1_Pine-TB composite.

## 4. Conclusions

Composite biosorbents based on bentonite clay and wood sawdust are able to remove Cu and Ni metal ions from aqueous solution. The adsorption capacities vary slightly from metal to metal and depend mainly on the type of wood used. In addition, acid modification of wood enhances the adsorption capacity due to increased surface area and pore volume. A maximum sorption capacity for biosorbent based on Taganskoe deposit Ca-bentonite and pine wood sawdust t_1_Pine-TB was found.

The initial rate of adsorption of heavy metals on biosorbents is very high followed by a slower rate indicating entry of the metal ions into the interior of the adsorbent particles. For both Cu and Ni ions removal, the adsorption process was well described by the pseudo-second order kinetic model which gives a better view of the suitable rate processes. The pine sawdust with bentonite clay sorbent revealed a good adsorption characteristic that makes it a good candidate as an inexpensive and effective adsorbent that has a high mechanical stability for heavy metal removal.

## Data Availability

The data presented in this study are available on request from the corresponding author.

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
