# Peer review of "Evaluation of Effective Composite Biosorbents Based on Wood Sawdust and Natural Clay for Heavy Metals Removal from Water"

_materials, 2023, doi:10.3390/ma16155322_

Round 1
Reviewer 1 Report
In this manuscript, authors combined the use of wood sawdust and bentonite for the heavy metals removal from water. Comprehensive characterizations have been performed. In general, it is an interesting work and the manuscript is well organized. However, there are still some issues to be addressed. A moderate revision is suggested before its acceptance.
1. The background should be simplified in abstract section.
2. More solid data should be added into the abstract section.
3. The normal keywords such as SEM, FTIR and XRD can be removed.
4. The different water pollution and treatments should be introduced with supporting articles, such as fluoride (New Journal of Chemistry 46, 490-497, 2022; Polymers 14 (24), 5417, 2022); dye; etc.
5. The last paragraph in introduction can be shortened to briefly introduce the novelty, strategy, method and important results.
6. More details on the raw materials should be provided, such as the purity.
7. One scheme to show the experimental procedure is suggested for better understanding of this work to readers.
8. It is better to make a table in the main manuscript to compare the heavy metal removal performance with other reports, such as: Cd; Cu2+( New Journal of Chemistry 47, 5639-5649, 2023); uranium; etc.
9. Three-line tables should be applied for a better scientific expression.
10. Some figures should be modified with a better readability, especially the quite small texts.
11. There are still some typos and grammar issues in the manuscript. Authors should carefully recheck the whole manuscript.
Author Response
REPLY TO REVIEWER 1
We wish to thank you for the evaluation and comments about our manuscript.
We have modified the manuscript taking into account your comments/suggestions.
All changes have been written in red in the manuscript.
Please find below our answers to your comments:
Point 1. The background should be simplified in abstract section.
Response 1. As suggested from the reviewer the background of the abstract section was simplified and more solid data were added as described in point 2
Point 2 . More solid data should be added into the abstract section.
Response 1 and Response 2. As suggested from the reviewer the background of the abstract section was simplified and more solid data were added as follow:
“Bentonitic clay and wood sawdust are natural materials widely available in nature at low cost with high heavy metal sorption properties that, in this work, were combined to achieve an effective composite biosorbent with high sorption properties and enhanced mechanical stability. Pine, aspen and birch wood sawdust, different bentonite clays and different sawdust modification methods (H3PO4 or HCl) have been used for preparing new composite biosorbents. A mixture of wood sawdust and bentonite in a ratio of 2:1 was used. All materials were characterized by using X-ray diffraction (XRD), Fourier transform infrared spectroscopy (FTIR) and scanning electron microscope (SEM) methods and tested for Cu and Ni ions removal from water. The adsorption process for all composite biosorbents was well described from a pseudo-second order kinetic model (R2 > 0.9999) with a very high initial adsorption rate of Cu and Ni ions and a maximum uptake recorded within 2 h. The results have shown that the adsorption capacity depends mainly on the kind of wood and with the acid treatment of wood that enhances the adsorption capacity. At a concentration of 50 mg/L, the biosorbent prepared using birch wood sawdust showed the worst performance, removing only almost 30% of Cu and Ni ions, while aspen wood sawdust improves the adsorption of Cu (88.6%) and Ni (52.4%) ions. Finally, composite biosorbent with pine wood sawdust showed the best adsorption behavior with an efficiency removal of 98.2 and 96.3% of Cu and Ni ions respectively, making it a good candidate as inexpensive and effective biosorbents for heavy metals removal.”
Point 3. The normal keywords such as SEM, FTIR and XRD can be removed.
Response 3. The keywords suggested from the reviewer were removed from the manuscript.
Point 4. The different water pollution and treatments should be introduced with supporting articles, such as fluoride (New Journal of Chemistry 46, 490-497, 2022; Polymers 14 (24), 5417, 2022); dye; etc.
Response 4. As suggested from the reviewer the required information were added in the text in the introduction section and in the references section as follow (consequently, all the other numbering of references were also changed):
“Many industries, such as automotive, metal finishing, electroplating, mining, electric cable manufacturing, tannery, steel, textile, and petroleum refining and rubber processing industries, discharge into the ecosystems and water bodies different concentrations of organic and inorganic pollutants such as dyes [1], pesticides [2], fluorides [3,4] and heavy metals like copper, lead, mercury, cadmium, zinc, uranium [5] , tungsten [6], cobalt and nickel [7]. Cu and Ni ions are the most common heavy metals in the above wastewaters [8,9].”
- Al-Tohamy, R.; Ali, S.S; Li, F.; Okasha, K.M.; Yehia A.-G. Mahmoud, Y.A-G.; Tamer Elsamahy, T.; Jiao, H.; Yinyi Fu, Y.; Sun, J. A critical review on the treatment of dye-containing wastewater: Ecotoxicological and health concerns of textile dyes and possible remediation approaches for environmental safety. Ecotoxicol. Environ. Saf. 2022, 231, 113160-113177. https://doi.org/10.1016/j.ecoenv.2021.113160.
- Syafrudin, M.; Kristanti, R.A.; Yuniarto, A.; Hadibarata, T.; Rhee, J.; Al-onazi, W.A.; Algarni, T.S.; Almarri, A.H.; Al-Mohaimeed, A.M. Pesticides in drinking water-a review. Int. J. Environ. Res. Public Health 2021, 18, 468-483. https://doi.org/10.3390/ijerph18020468.
- Jian, S.; Cheng, Y.; Ma, X.; Guo, H.; Hu, J.; Zhang, K.; Jiang, S.; Yang W.; Duan G. Excellent fluoride removal performance by electrospun La–Mn bimetal oxide nanofibers. New J. Chem. 2022, 46, 490. https://doi.org/10.1039/d1nj04976crsc.li/njc.
- Jian, S.; Chen, Y.; Shi, F.; Liu, Y.; Jiang, W.; Hu, J.; Han, X.; Jiang, S.; Yang, W. Template-free synthesis of magnetic La-Mn-Fe tri-metal oxide nanofibers for efficient fluoride remediation: kinetics, isotherms, thermodynamics and reusability. Polymers 2022, 14, 5417. https://doi.org/10.3390/polym14245417.
- Hassanin, M.A.; Negm, S.H.; Youssef, M.A.; Sakr, A.K.; Mira, H.I.; Mohammaden, T.F.; Al-Otaibi, J.S.; Hanfi, M.Y.; Sayyed, M.I.; Cheira, M.F. Sustainable remedy waste to generate SiO2 functionalized on graphene oxide for removal of U(VI) ions. Sustainability 2022, 14, 2699. https://doi.org/10.3390/su14052699.
- Elbshary, R.E.; Gouda, A.A.; El Sheikh, R.; Alqahtani, M.S.; Hanfi, M.Y.; Atia, B.M.; Sakr, A.K.; Gado, M.A. Recovery of W(VI) from wolframite ore using new synthetic schiff base derivative. Int. J. Mol. Sci. 2023, 24, 7423. https://doi.org/10.3390/ ijms24087423.
- El Hajam, M.; Kandri, N.I.; Plavan, G.-I.; Harrath, A.H.; Mansour, L.; Boufahja, F.; Zerouale, A. Pb2+ ions adsorption onto raw and chemically activated Dibetou sawdust: Application of experimental designs. J. King Saud Univ. Sci. 2020, 32, 2176-2189. https://doi.org/10.1016/j.jksus.2020.02.027.
- Kazeminezhad, I.; Mosivand S. Elimination of copper and nickel from wastewater by electrooxidation method. J. Magn. Magn. Mater. 2017, 422, 84–92. http://dx.doi.org/10.1016/j.jmmm.2016.08.049.
- Zhang, Y.; Wang K.; Duan, G.; Chen, Y.; Liu, K.; Hou H. Efficient removal of high- or low-concentration copper ions using diethylenetriamine-grafted electrospun polyacrylonitrile fibers. New J. Chem. 2023, 47, 5639.https://10.1039/d2nj05789arsc.li/njc.
Point 5. The last paragraph in introduction can be shortened to briefly introduce the novelty, strategy, method and important results.
Response 5. As properly noted from the reviewer the last paragraph in the introduction section is very long. For this reason we shortened this paragraph as follow:
“This work aims to combine for the first time the natural adsorption properties of wood sawdust and bentonitic clay to achieve an effective heavy metals sorption material with high mechanical stability and good sorption properties from water treatment. In detail pine, aspen and birch wood sawdust, different bentonite clays and different sawdust modification methods were used to prepare composite biosorbents and their sorption properties were compared through their application for removal of Cu and Ni ions from water in batch experiments. Composite biosorbents characterization by using of X-ray diffraction (XRD), Fourier transform infrared spectroscopy (FTIR), scanning electron microscope (SEM), and energy dispersive X-ray (EDX) methods was made. To investigate the adsorption mechanism of metal ions onto the sorbent, four kinetic models were analyzed: first order reaction model, pseudo-first order equation, second order reaction model, and pseudo-second order reaction model.”
Point 6. More details on the raw materials should be provided, such as the purity.
Response 6. As suggested from the reviewer in the paragraph 2.1 Materials, the purity of all materials was provided.
Point 7. One scheme to show the experimental procedure is suggested for better understanding of this work to readers.
Response 7. As correctly suggested, a new scheme (Scheme 1) was prepared and added into the manuscript at the beginning of Results section and before Table 2 (paragraph 3.1). Moreover the scheme was also mentioned into the text.
Point 8. It is better to make a table in the main manuscript to compare the heavy metal removal performance with other reports, such as: Cd; Cu2+( New Journal of Chemistry 47, 5639-5649, 2023); uranium; etc.
Response 8. Taking into consideration the suggestion of the reviewer we modified table 5 comparing the heavy metal removal performance of our work with other reports. However, we prefer to compare the adsorption performance of similar composite biosorbents in order to have a truthful comparison, while the interesting work suggested (New Journal of Chemistry 47, 5639-5649, 2023) was added in the introduction section.
Moreover we added the following references in the reference section:
- Bhattacharyya, K.G.; Sen Gupta, S. Pb(II) uptake by kaolinite and montmorillonite in aqueous medium: influence of acid activation of the clays. Colloids Surf. A Physicochem. Eng. Asp. 2006,191–200. https://doi.org/10.1016/j.colsurfa.2005.11.060.
- Turan, N.G.; Elevli, S.; Mesci, B. Adsorption of copper and zinc ions on illite: determination of the optimal conditions by the statistical design of experiments. Appl. Clay Sci. 2011, 52,392–399, https://doi.org/10.1016/j.clay.2011.04.010.
- Annadurai, G.; Juang, R.-S.; Lee, D. Adsorption of heavy metals from water using banana and orange peels. Water Sci. Technol. 2003, 47, 185–190. https://doi.org/10.2166/wst.2003.0049.
- Stefan Demcaka, S.; Balintova, M.; Demcakova, M.; Csach, K.; Inga Zinicovscaia, I.; Nikita Yushin, N.; Frontasyeva, M. Effect of alkaline treatment of wooden sawdust for the removal of heavy metals from aquatic environments. Desalin. Water Treat. 2019, 155,207–215. https://doi: 10.5004/dwt.2019.24053.
- Du, W.; Cui, S.; Xing Fang, X.; Wang, Q.; Liu, G. Adsorption of Cd(II), Cu(II), and Zn(II) by granules prepared using sludge from a drinking water purification plant. J. Environ. Chem. Eng. 2020, 8,104530. https://doi.org/10.1016/j.jece.2020.104530.
- Siswoyo, E.; Mihara, Y.; Tanaka, S. Determination of key components and adsorption capacity of a low cost adsorbent based on sludge of drinking water treatment plant to adsorb cadmium ion in water. Appl. Clay Sci. 2014, 97–98,146–152. https:// doi.org/10.1016/j.clay.2014.05.024.
Point 9. Three-line tables should be applied for a better scientific expression.
Response 9. As suggested from the reviewer, for a better scientific expression, three-line tables were applied for table 2 and table 3.
Point 10. Some figures should be modified with a better readability, especially the quite small texts.
Response 10. In order to have a better readability Fig.s 6 and 8 have been modified while Fig.s 7 and 9 have been enlarged
Point 11. There are still some typos and grammar issues in the manuscript. Authors should carefully recheck the whole manuscript.
Response 11. As suggested from the reviewer the manuscript was rechecked and all corrections were written in red in the text. For instance we changed “biosorbent composites” to “composite biosorbents” in the whole manuscript (title included).

Reviewer 2 Report
All comments and suggestions are in word file

Minor editing of English language required
Author Response
REPLY TO REVIEWER 2
We wish to thank you for the evaluation and comments about our manuscript.
We have modified the manuscript taking into account your comments/suggestions.
All changes have been written in red in the manuscript.
Please find below our answers to your comments:
Point 1. There are minor grammatical mistakes. Please check the manuscript for grammar and English carefully.
Response 1. We have thoroughly checked into the entire manuscript and corrected grammatical errors.
Point 2. The abstract is well presented.
Response 2. We have slightly modified to further improve the abstract taking into consideration the suggestion of the other reviewer on the abstract section
Point 3. The literature survey in the introduction section can be enriched by adding some recent references related to different natural sorbents, such as: Sustainability 2022, 14(5), 2699 doi:10.3390/su14052699; Int. J. Mol. Sci. 2023, 24(8), 7423 doi:10.3390/ijms24087423.
Response 3. As suggested from the reviewer, we added other recent references in the introduction and consequently we have updated the bibliography.
Point 4. The experimental section is well organized.
Response 4. We thank you the reviewer for this comment
Point 5. L 180, for kinetics study
Response 5.The suggested change in line L180 has been made
Point 6. Results and Discussion section: What is the pH of the biosorbent?
Response 6. We forgot to add this information into the text, thus the following sentence was added in Results section (paragraph 3.2): “Biosorbents pH was controlled during the preparation process giving pH values in water for all the prepared composite biosorbents around 6.”
Point 7. Characterization of XPS will be interesting.
Response 7. Even if the suggested information will be interesting, unfortunately we do not have availability to this facility in a short time.
Point 8. The effect of different cations should be studied.
Response 8. As suggested we studied other analytes and the following sentences were added into the text at the end of Results section (paragraph 3.3):
“We also verified the behaviour on a different heavy metal ion by monitoring the sorption capacity of t1Pine-TB biosorbent for chromium(VI) ions. The sorption capacity was about 7.5 mg/g. A preliminary evaluation on the capability of the biosorbent to extract oil products was also made. The sorption capacity in this last case was about 3.6 mg/g and it confirms the large sorption potential of the developed t1Pine-TB composite.”
Point 9. The non-linear kinetics fitting of the data is strongly suggested.
Response 9. For the processing of kinetic data, we used well-known models of the first order, pseudo-first order, second order and pseudo-second order (Figure 7, 9). We evaluated the applicability of models for the possibility of linear approximation of kinetic dependences using first-order, pseudo-first-order, second-order, and pseudo-second-order equations. This estimation method was chosen as the most widely used in the description of sorption processes and allows a fairly good description of the sorption kinetics.
Following the reviewer suggestion, nonlinear kinetics were determined, and the following sentences were added into the text in Results section (paragraph 3.3):
page 14. “Although, linear approximation of kinetic dependences was chosen as the most widely used in the description of sorption processes and allowed a fairly good description of the sorption kinetics, the nonlinear pseudo-second order was also done by using the following equation (12):
q_t=(K_2 q_e^2 t)/(〖1+K_2 q〗_e t) (12)
where K2 is the pseudo-second order constant and qt and qe are the amount (mg) of metal ion adsorbed per unit mass of the adsorbent (g) at time t and at equilibrium respectively. For experiments with Cu ions (C0=50 mg/L) nonlinear Equation (12) gave values of qe and R2 equal to 4.92 mg/g and 0.999 respectively while at C0=10 mg/L values of qe and R2 equal to 1.06 mg/g and 0.800 respectively.”
page 16. “Also for Ni ions nonlinear pseudo-second order Equation (12) was tested, giving similar results to those of Cu ions and confirming the convenience of the linearization approach.”
Point 10. Captions of fig. 7 and 9 should be revised.
Response 10. As suggested from the reviewer captions of Fig.7 and 9 has been modified as follows:
Fig. 7 Linear kinetic models of Cu ions (10 and 50 mg/L) adsorption on t1Pine-TB: first order (a), pseudo-first order (b), second order (c), and pseudo-second order (d) models
Fig. 9 Linear kinetic models of Cu ions (10 and 50 mg/L) adsorption on t1Pine-TB: first order (a), pseudo-first order (b), second order (c), and pseudo-second order (d) models

Round 2
Reviewer 1 Report
All issues have been well addressed. An acceptance is suggested.
Reviewer 2 Report
Accept in present form